# Molecular Pathology of Thyroid Tumors: Essential Points to Comprehend Regarding the Latest WHO Classification

**DOI:** 10.3390/biomedicines12040712

**Published:** 2024-03-22

**Authors:** Tomohiro Chiba

**Affiliations:** 1Department of Cytology, Cancer Institute Hospital, Japanese Foundation for Cancer Research, 3-8-31 Ariake, Koto-ku, Tokyo 135-8550, Japan; tomohiro.chiba@jfcr.or.jp; Tel.: +81-3-3520-0111; Fax: +81-3-3570-0558; 2Department of Pathology, Cancer Institute Hospital, Japanese Foundation for Cancer Research, 3-8-31 Ariake, Koto-ku, Tokyo 135-8550, Japan

**Keywords:** thyroid cancer, genomic alterations, pathological diagnosis, WHO classification

## Abstract

In 2022, the new WHO Classification of Endocrine and Neuroendocrine Tumors, Fifth Edition (beta version) (WHO 5th), was published. Large-scale genomic analyses such as The Cancer Genome Atlas (TCGA) have revealed the importance of understanding the molecular genetics of thyroid tumors. Consequently, the WHO 5th was fundamentally revised, resulting in a systematic classification based on the cell of origin of tumors and their clinical risk. This paper outlines the following critical points of the WHO 5th. 1. Genetic mutations in follicular cell-derived neoplasms (FDNs) highlight the role of mutations in the MAP kinase pathway, including *RET*, *RAS*, and *BRAF*, as drivers of carcinogenesis. Differentiated thyroid cancers such as follicular thyroid carcinoma (FTC) and papillary thyroid carcinoma (PTC) have specific genetic alterations that correlate with morphological classifications: *RAS*-like tumors (RLTs) and *BRAF* p.V600E-like tumors (BLTs), respectively. 2. The framework for benign lesions has been revised. The WHO 5th introduces a new category: “developmental abnormalities”. Benign FDNs comprise “thyroid follicular nodular disease”, follicular thyroid adenoma (FTA), FTA with papillary architecture, and oncocytic adenoma (OA). “Hürthle cell adenoma/carcinoma” is renamed oncocytic adenoma/carcinoma of the thyroid (OA/OCA), which can be distinguished from FTA/FTC by its unique genetic background. 3. Low-risk tumors include NIFTP, TT-UMP, and HTT, and they have an extremely low malignant potential or an uncertain malignant potential. 4. PTC histological variants are reclassified as “subtypes” in the WHO 5th. 5. The concept of high-grade carcinomas is introduced, encompassing poorly differentiated thyroid carcinoma (PDTC), differentiated high-grade thyroid carcinoma (DHGTC), and high-grade medullary thyroid carcinoma (MTC). 6. Squamous cell carcinoma is included in anaplastic thyroid carcinoma (ATC) in the WHO 5th due to their shared genetic and prognostic features. 7. Other miscellaneous tumors are categorized as salivary-gland-type carcinomas of the thyroid, thyroid tumors of uncertain histogenesis, thymic tumors within the thyroid, and embryonal thyroid neoplasms. The WHO 5th thus emphasizes the importance of classifying tumors based on both genetic abnormalities and histomorphology. This approach aids in achieving accurate pathological diagnosis and facilitates the early selection of appropriate treatment options, including molecular targeted therapies.

## 1. Introduction

Thyroid cancer diagnoses, particularly in women, are on the rise. Improved detection methods have likely contributed to this trend, although environmental factors are also under investigation. Despite the increase in diagnoses, mortality rates remain low. Thyroid tumors are classified by listing the major histologic types: follicular thyroid adenoma (FTA), a benign tumor; papillary thyroid carcinoma (PTC); follicular thyroid carcinoma (FTC); anaplastic thyroid carcinoma (ATC); and medullary thyroid carcinoma (MTC) [1,2]. The results of large-scale genome analyses, including TCGA (The Cancer Genome Atlas), have accumulated. The fourth edition of the WHO Classification of Endocrine Tumors (WHO 4th), published in 2017, included information on the driver genes associated with thyroid cancer [2]. WHO released a beta version of the new classification of endocrine and neuroendocrine tumors (WHO 5th) in 2022 [3,4]. Continuing the trend of the WHO 4th and taking tumor-specific molecular genetic information into account, the WHO 5th has been fundamentally revised. It now includes a systematic classification based on the cell of origin and the clinical risk (Table 1). The main categories are follicular cell-derived neoplasms (FDNs) and parafollicular cell (C cell)-derived tumors, with FDNs further divided into three classes: benign, low-risk, and malignant (Table 1). The only C cell-derived tumor is the malignant MTC, which encompasses low-grade and high-grade subcategories. Other tumors are also re-classified as salivary-gland-type carcinomas of the thyroid, thyroid tumors of uncertain histogenesis, thymic tumors within the thyroid, and embryonal thyroid neoplasms. This paper outlines critical points that enable us to better understand the WHO 5th.

## 2. Essential Points to Comprehend the Latest WHO Classification

### 2.1. Genetic Mutations in Follicular Cell-Derived Tumors (FDNs)

In FDNs, genetic abnormalities in the MAP kinase pathway, starting from receptor-type tyrosine kinases such as *RET*, *RAS*, and *BRAF*, are the drivers of carcinogenesis [2,3,4,5]. In differentiated thyroid cancers such as PTC and FTC, a limited number of mutations such as *BRAF* p.V600E, *RET* translocations (*CCDC6::RET*, *NCOA4::RET*, etc.), *H/K/NRAS* mutations such as *NRAS* p.Q61R, and *PAX8::PPARG* translocation are often found, and these are mutually exclusive (Figure 1).

*BRAF* p.V600E mutation, along with *RET* and *NTRK* rearrangements, have been identified as primary genetic aberrations in PTC [2,3,4]. Notably, *BRAF* p.V600E exhibits a high specificity for PTC. Their detection, even in minute quantities within fine-needle aspiration cytology specimens, plays a pivotal role in diagnosing PTC [6]. Additionally, *RAS* point mutations have been documented in follicular PTC and FTC. It is worth noting that *RAS* mutations have also been observed in FTA and benign thyroid nodules. Furthermore, nodules harboring *RAS* mutations have been associated with accelerated growth rates [7]. Unlike the well-known *BRAF* p.V600E mutation, other alterations in the *BRAF* gene, such as p.K601 mutations and gene rearrangements, have been detected in FTC as well as benign follicular tumors [2,3,4].

The findings from a comprehensive thyroid cancer genome analysis [8] have demonstrated a strong correlation between driver gene types and morphological classifications. Through genetic alteration pattern analysis, the TCGA study introduced the BRAF-RAS score, delineating two distinct subgroups within PTC: *BRAF* p.V600E-like tumors (BLTs) and *RAS*-like tumors (RLTs). Moreover, compiled evidence from several next-generation sequence (NGS) analyses have demonstrated that this dichotomous classification is not only applicable to PTC, but also extends to FTA and FTC, with BLTs displaying a classic PTC morphology and RLTs encompassing FTA, FTC, and follicular PTC [Korea].

The driver gene mutations of the typical FTC are *RAS* mutations or *PAX8::PPARG* translocation (PAX8::PPARG fusion protein), and FTC grows expansively with a retained follicular structure and forms a fibrous capsule (Figure 1). FTC expresses iodine metabolism-related and hormone-related genes, showing thyroidal differentiation. Such tumors are called “RAS-like tumors (RLTs)” with regard to the genetic mutations.

On the other hand, the typical PTC has *BRAF* p.V600E mutations and *RET* fusion genes and grows invasively with a papillary structure and characteristic nuclear atypia (e.g., ground glass-like chromatin, nuclear grooves, and intranuclear cytoplasmic inclusions). Such tumors are referred to as “*BRAF* p.V600E-like tumors (BLTs)”. BLTs have a lower iodine metabolic capacity and lower hormone differentiation compared to RLTs. The *BRAF* p.K601E mutation is found in RLTs. Some tumors are difficult to classify as BLTs or RLTs (non-*BRAF*/non-*RAS* tumors: NBNRs).

### 2.2. Revision of the Framework for Benign Lesions

The fourth edition of WHO (WHO 4th) listed only FTA and Hürthle cell adenoma as benign lesions. The WHO 5th adopted a new category, namely “developmental abnormalities”, to include pathologies such as thyroglossal duct cysts; it also added “thyroid follicular nodular disease (TFND [or multinodular goiter: MNG])”, which often exhibits clonality [3,4,5], to the benign FDN classification (Table 1).

While most FTAs are RLTs, TFND has a cluster of genetic abnormalities in the thyroid-stimulating hormone receptor (TSHR) and its downstream pathway consisting of Gsα-adenylyl cyclase-protein kinase A (PKA). The most frequent mutation in TFND is the *TSHR* mutation (~70%). Other driver genes in TFND include *GNAS*, *EZH1*, *ZNF148*, and *SPOP* [9,10]. Carney complex (CNC) is an autosomal dominant tumor predisposition syndrome that affects endocrine glands, including the thyroid [3]. Up to 60% of CNC patients develop thyroid nodules. CNC is associated with pathological variations in *PRKAR1A* downstream of cAMP. DICER1 syndrome, caused by pathogenic variants in *DICER1*, is another autosomal dominant tumor predisposition syndrome that affects multiple organs, including the thyroid [3]. The most common manifestation is benign thyroid nodules (MNG), which occur in approximately 75% of females and 10–20% of males with *DICER1* pathogenic variants by the age of 40.

FTA with papillary architecture (FTAPA) is a benign non-invasive encapsulated FDN characterized by an intrafollicular papillary architecture; this lacks PTC-like nuclear features and is often associated with hyperthyroidism. Like TFND, mutations in *TSHR*, *GNAS*, *PRKAR1A*, and *EZH1* are common.

Hürthle cell adenoma (formerly called follicular adenoma, oxyphilic cell variant) is renamed oncocytic adenoma of the thyroid (OA) in the WHO 5th. Accordingly, the malignant counterpart of OA is renamed oncocytic carcinoma of the thyroid (OCA). The name “Hürthle cell tumor” is no longer used because it is inappropriate. OA/OCA were distinguished from FTA/FTC by their characteristic morphology, as well as their unique genetic background [11,12]. OA/OCA have a high frequency of gene mutations in the mitochondrial biosynthesis system, such as *ESRRA* and *PPARGC1A*, and have characteristic genetic abnormalities such as a near-haploid (or monoploid) karyotype. OA/OCA, on the other hand, have a low frequency of *RAS* mutations and *PAX8::PPARG* translocations, which are the major driver mutations of FN. Oxyphilic PTC, which generally has *BRAF* p.V600E-like genetic backgrounds, is not included in OCA.

The criteria for differentiating OA from OCA are the same as those for follicular tumors: the presence of capsular invasion and vascular invasion. Like the subclassification of FTC, OCA is subdivided into three subtypes: minimally invasive, encapsulated angioinvasive, and widely invasive OCA.

### 2.3. Low-Risk Tumors

“Low-risk tumors” in FDNs are a morphological and clinical intermediate between benign and malignant tumors. Low-risk tumors have the potential to metastasize but do so infrequently. Low-risk tumors comprise non-invasive follicular thyroid neoplasms with papillary-like nuclear features (NIFTPs), thyroid tumors of uncertain malignant potential (TT-UMPs), and hyalinizing trabecular tumors (HTTs).

NIFTPs are encapsulated FDNs with a follicular growth pattern and PTC-like nuclear features, lacking capsular and/or vascular invasion (Figure 2A–C). NIFTPs were previously diagnosed as an encapsulated follicular variant PTC and renamed due to their favorable prognosis [13]. The exclusion criteria for NIFTPs are (i) psammoma bodies, (ii) a mitotic count of >3/2 mm^2^, (iii) tumor necrosis, and (iv) the presence of genetic alterations including *BRAF* p.V600E, *RET* rearrangement, and *TERT* promoter mutation. In WHO 5th, NIFTPs now incorporate tumors smaller than 1 cm and oncocytic tumors, which were previously excluded.

TT-UMPs are defined as tumors of “questionable” capsular or vascular invasion; those without PTC-like nuclear atypia are follicular tumors of uncertain malignant potential (FT-UMPs), and those with PTC-like nuclear atypia are well-differentiated tumors of uncertain malignant potential (WDT-UMPs). These diagnostic terms should be carefully used after a thorough pathological investigation of the specimen. Driver gene mutations in TT-UMPs are diverse. While RAS-like mutations are the most common type, mutations in *EIF1AX* and *TSHR*, as well as *PAX8::PPARG* rearrangements, can also be detected.

HTTs were first reported by Carney et al. in 1987 as benign tumors (hyalinizing trabecular “adenoma”) and were later renamed after findings that were suggestive of malignancy, such as lymph node metastasis, were presented [14,15]. The presence of PTC-like nuclear atypia and the cell membrane staining of Ki-67 (MIB1), but not the nucleus, are key diagnostic features of HTTs (Figure 2D–I). HTTs commonly exhibit the translocations of *PAX8::GLIS3* (93%) or *PAX8::GLIS1* (7%) [16]. Neither *RAS* nor *BRAF* mutations have been detected in HTTs.

### 2.4. Subtypes of PTC

PTC subtypes have been described as “variants”, but to distinguish them from genetic variants, the term “subtype” is adopted in the WHO 5th (Table 2). The previous “cribriform morular variant PTC” is considered as “cribriform morular carcinoma” in tumors of uncertain histogenesis because they are associated with the constitutive activation of the WNT/β-catenin pathway, which can occur with familial adenomatous polyposis or sporadically. Among the follicular PTCs, those with wide invasive growth remain in the PTC subtype as infiltrative follicular PTC (ifPTC), while the invasive encapsulated follicular variant of papillary thyroid carcinoma (IEFVPTC) has become a unique classification. This is because the genetic background classifies most ifPTCs as BLTs and most IEFVPTCs as RLTs.

The subtypes with a poor prognosis include tall-cell PTC (tcPTC), hobnail PTC (hPTC), and columnar cell PTC (ccPTC). These predominantly comprise *BRAF* p.V600E mutations and often meet the diagnostic criteria for high-grade differentiated carcinomas, as described below. Solid/trabecular PTC (stPTC) also constitutes a slightly higher risk and has a higher frequency of *RET* rearrangements. tcPTC and hPTC are diagnosed when they represent more than 30% of all tumors, and stPTC when it represents more than half of all tumors. CDX2 is often immunohistochemically positive in ccPTC. The risk of diffuse sclerosing PTC is controversial, but it might constitute a higher risk [17].

### 2.5. High-Grade Thyroid Carcinomas

It has been repeatedly reported that some differentiated follicular cell-derived carcinoma cases, including FTC, PTC, and oncocytic carcinoma (OCA), and some MTC cases have a poor prognosis [3,4,5]. The WHO 5th introduced the concept of high-grade carcinomas, encompassing high-grade follicular cell-derived non-anaplastic thyroid carcinoma and high-grade MTC. The former category includes PDTC and a newly proposed histological classification termed “differentiated high-grade thyroid carcinoma (DHGTC)” (Table 3).

Poorly differentiated thyroid carcinoma (PDTC) has clinicopathological characteristics that are intermediate between well-differentiated follicular cell-derived carcinoma with excellent prognosis and anaplastic thyroid carcinoma (ATC). PDTC is diagnosed based on the Turin consensus criteria [18]: (i) the presence of a solid/trabecular/insular growth pattern, (ii) an absence of the conventional nuclear features of papillary carcinoma, and (iii) the presence of at least one of the following: convoluted nuclei, increased mitotic counts (≥3 per 2 mm^2^), and tumor necrosis.

In well-differentiated follicular cell-derived carcinomas, there are high-risk cases comparable to PDTC [19]. These cases are named “DHGTC” in the WHO 5th and can be morphologically differentiated by increased mitotic counts (≥5 fissions/2 mm^2^) or/and tumor necrosis in authentic differentiated thyroid carcinomas including FTC, PTC, and OCA (Figure 3).

In PDTC and DHGTC, *RAS* and *BRAF* mutations are detected at a frequency similar to that in well-differentiated carcinomas. The *TP53*, *CDKN2A*, *PIK3CA*, and *AKT1* mutations are high-risk mutations related to a poor prognosis and malignant transformation. *TERT* promoter mutations (C228T and C250T) are also high-risk genetic alterations. Several studies have suggested the prognostic importance of the co-existence of *BRAF* p.V600E and *TERT* promoter mutations in differentiated thyroid cancer [20,21]. This combination of *BRAF* p.V600E and *TERT* promoter mutations is called a “genetic duet”. Recent studies have further recognized the importance of the regulatory single-nucleotide polymorphism (rSNP) rs2853669 in the *TERT* promoter as an additional predictor of risk for PTC [22,23].

The concept of high-grade carcinoma was also introduced for MTC, a calcitonin-producing C cell-derived carcinoma. MTC exhibits neuroendocrine differentiation and is a primary neuroendocrine tumor/carcinoma (NET/NEC) of the thyroid gland. Like NET/NEC in other organs, the prognosis for MTC varies significantly from case to case. The proliferative activities of the tumor can stratify the long-term risk of MTC [24,25]. The WHO 5th recommended a two-tier risk assessment system based on the tumor’s proliferative activity and necrosis [4]: (i) mitotic counts, ≥5 cells/2 mm^2^; (ii) Ki67 labeling index, ≥5%; and (iii) presence of tumor necrosis (Table 3).

Most cases of hereditary MTC and about half of sporadic MTC cases have *RET* mutations. The second most frequent mutations in sporadic MTC are *RAS* mutations. The clinical risk varies depending on the type of *RET* mutation, with the frequent *RET* p.M918T mutation being that with the highest risk [3,4,5].

### 2.6. Changes in the Definition of Anaplastic Thyroid Carcinoma (ATC)

Squamous cell carcinoma (SCC) without a well-differentiated thyroid carcinoma component has been considered a unique histologic type. The WHO 5th edition has incorporated SCC into ATC because SCC of the thyroid generally shows *BRAF* p.V600E mutations (87%) and is immunohistologically positive for the follicular cell markers PAX8 (91%) and TTF1 (38%) [3,4,5]. SCC of the thyroid also exhibits a poor prognosis comparable to that of other ATCs. The squamous metaplasia of follicular cells and the squamous differentiation of PTC should not be mistaken for ATC. The direct invasion of primary SCC of the head and neck region should also be excluded.

### 2.7. Miscellaneous Thyroid Tumors

Miscellaneous tumors were classified into four categories based on their cellular origin or differentiation: (i) salivary-gland-type carcinomas of the thyroid, (ii) thymic tumors within the thyroid, (iii) thyroid tumors of uncertain histogenesis, and (iv) embryonal thyroid neoplasms (Table 1).

Salivary-gland-type tumors include mucoepidermoid carcinoma of the thyroid (MEC) and secretory carcinoma of the salivary gland type (SC). MEC consists predominantly of epidermoid cells mixed with a smaller number of mucocytes. Two theories exist regarding the origin of MEC: one posits derivation from ectopic salivary glands, while the other suggests derivation from a solid cell nest within the thyroid gland. In some MEC cases, the *CRTC1::MAML2* fusion gene, characteristic of salivary gland MEC, is detected. SC cases commonly involve the *ETV6::NTRK3* fusion gene.

Intrathyroid thymic tumors include thymoma, intrathyroidal thymic carcinoma (ITC), and spindle cell tumor with thymus-like differentiation (SETTLE). The WHO 4th has remained the same for these classifications.

Sclerosing mucoepidermoid carcinoma with eosinophilia (SMECE) and cribriform morular thyroid carcinoma (CMTC) were classified as tumors of uncertain histogenesis. SMECE was considered a subtype of MEC, but became an independent histologic type due to the absence of the *MAML2* fusion gene and the presence of Hashimoto’s disease in the background. CMTC, which was a subtype of PTC in the WHO 4th, is related to familial adenomatous polyposis (FAP) and genetic abnormalities in the β-catenin system such as *APC*. CMTC became independent because it does not show apparent follicular cell differentiation, namely the lack of thyroglobulin production.

Thyroblastoma has been introduced in a new category of embryonal thyroid neoplasms. Thyroblastoma is a highly aggressive tumor, consisting of primitive thyroid-like follicular cells, a primitive small cell component, and mesenchymal stroma. Most thyroblastoma cases have been classified as malignant thyroid teratoma or carcinosarcoma. *DICER1* somatic mutations are common.

It is noteworthy that lymphomas and mesenchymal tumors have been removed from the specific classification of thyroid tumors because they are now grouped together with other endocrine organs.

## 3. Diagnostic Procedure According to WHO 5th

The WHO 5th simplified the diagnosis of thyroid cancer by dividing it into three steps, as illustrated in Figure 4. The first step involves considering the cell of origin or the cellular differentiation of the tumor cells. The second steps involves an assessment of gene mutations, while the third entails a detailed examination of various histomorphological features, such as capsular/vascular invasion, mitosis, and tumor necrosis.

Following the diagnostic algorithm enables the efficient differential diagnosis of thyroid tumors. In facilities with access to genetic analysis, a preliminary diagnosis can be made without relying solely on histomorphological analyses. Conversely, in settings where genetic analysis is not available, it is still possible to narrow down the types of driver genes to some extent through histomorphological analyses. The 5th edition of the WHO aims to minimize interdiagnostic variability and promote more accurate and timely diagnoses.

### 3.1. Consideration of the Cell-of-Origin or Cellular Differentiation

Thyroid tumor diagnosis primarily relies on determining the cell of origin or tumor differentiation, which is evaluated through histomorphological observation and differentiation markers. For follicular cell differentiation, immunohistological markers such as TTF1, PAX8, and Thyroglobulin are utilized. Additional differentiation markers include calcitonin for C cell differentiation, CD5 for thymus-like differentiation, and SALL4 for embryonal features (Figure 4). Gene expression analysis is also capable of identifying cell differentiation, with current NGS-based thyroid cancer comprehensive panel tests utilizing gene expression patterns for this purpose. Additionally, these tests can directly detect unique driver gene mutations that are closely associated with diagnosis, including *APC*, *CRTC1::MAML2*, and *ETV6::NTRK3*.

### 3.2. Genetic-Morphologic Classification of the FDNs into Subcategories

FDNs, the most frequent thyroid tumors, are generally classified into five types according to their genetic alterations: (i) oncocytic tumors, (ii) well-differentiated tumors such as TFND, (iii) RLTs, (iv) BLTs, and (v) high-grade carcinomas and ATC. This classification can be performed by gene expression analysis. This classification can be readily achieved using NGS-based genetic analysis. However, the strong correlation between driver genes and morphological features in thyroid cancer allows for a morphological observation-based classification as well. Immunostaining for mitochondria aids in differentiating oncocytic tumors. The diagnosis of oncocytic tumors requires the presence of oncocytes in more than 75% of tumors.

PDTC and ATC typically display solid growth patterns, whereas DHGTC can manifest various growth patterns. DHGTC is diagnosed in FTC, PTC, and OCA based on the presence of increased mitotic figures and/or tumor necrosis. It should be noted that tumors showing solid growth patterns with PTC-like nuclear features are diagnosed as the PTC subtype “stPTC”.

### 3.3. Detailed Histological Examination for the Diagnosis

Further detailed histomorphological evaluation is necessary to confirm the diagnosis of FDNs. Oncocytic tumors and RLTs typically present as encapsulated nodular lesions. The evaluation of capsular and vascular invasion is crucial in determining their benign or malignant status. In RLTs, the diagnosis is determined by assessing the presence or absence of PTC-like nuclear features (nuclear score 2–3 for presence, nuclear score 0–1 for absence) and capsular and vascular invasion. A diagnosis of FTA is established when there are no PTC-like nuclear features and no capsular or vascular invasion. Conversely, a diagnosis of FTC is made when there are no PTC-like nuclear features, but capsular or vascular invasion is present. This indicates that the presence of a *RAS* mutation alone is insufficient to determine malignancy. When PTC-like nuclear features are present without capsular or vascular invasion, the tumor is classified as a NIFTP; meanwhile, if PTC-like nuclear features are present with capsular or vascular invasion, the tumor is categorized as IEFVPTC. When the capsular or vascular invasion is “questionable”, the diagnostic term “uncertain malignant potential (UMP)” is used. If the tumor shows PTC-like nuclear features, the diagnosis is well-differentiated tumor-UMP (WDT-UMP); otherwise, it is follicular tumor-UMP (FT-UMP).

High-grade carcinoma and ATC are associated with high-risk gene mutations such as *TERT* promoter mutations, *TP53* mutations, and *CDKN2A/2B* loss, which can be detected through genetic analysis. Tumors harboring these high-risk mutations exhibit morphological features such as brisk mitotic activity and tumor necrosis. These characteristics are utilized to identify high-grade carcinomas morphologically. The diagnosis of high-grade carcinomas is crucial for the early initiation of molecular targeted therapy because a variety of drugs such as Selpercatinib, Entrectinib, Dabrafenib, and Trametinib are currently available for radioactive iodine-resistant thyroid carcinomas.

## 4. Conclusions

In the WHO 5th edition, the classification was revamped to a systematic approach based on the cell of origin, clinical risk, and genetic alterations. In follicular cell-derived tumors, there exists a close relationship between driver genes and morphological features, enabling the estimation of driver genes to some extent through morphological observation. Conversely, if the driver gene can be identified via genetic testing, it can provide additional confirmation for histological diagnosis. This classification system facilitates the standardization of pathological diagnosis for thyroid tumors, enabling early intervention for high-grade tumors. Understanding genetic alterations has become essential not only in the pathological diagnosis of thyroid tumors, but also in the selection of treatment options.

## Figures and Tables

**Figure 1 biomedicines-12-00712-f001:**
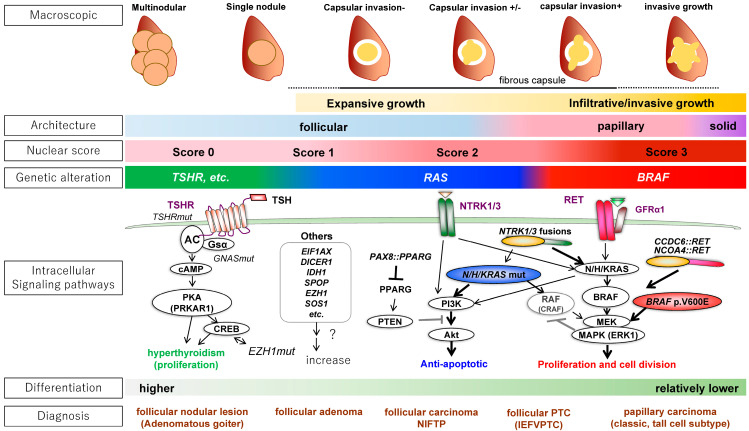
Morphology, genetic abnormalities, and intracellular signaling pathways of differentiated thyroid tumors. The three most important gross characteristics of thyroid tumors are (i) single or multiple nodules, (ii) capsule formation, and (iii) the presence or absence of invasive growth. A lesion with a single nodule and a thick fibrous capsule indicates a neoplastic lesion that has expansively grown over time. Multinodular lesions are more likely to be non-neoplastic. Lesions with irregular margins are suspected of infiltrative growth and may be malignant. The most important histological findings are the histological architecture (follicular, papillary, or solid) and papillary carcinoma (PTC)-like nuclear features (“nuclear score”: nuclear enlargement, glassy chromatin, and irregular nuclear shape [nuclear grooves and pseudo-inclusions]), with one point given for each [Total 0–3 points]). Regarding genetic mutations, well-differentiated tumors such as thyroid follicular nodular disease (or functional nodules) show abnormalities in the TSHR to GNAS/cAMP/PKA pathway. Differentiated tumors can be classified into *RAS*-like tumors (RLTs) and *BRAF* p.V600E-like tumors (BLTs). Many of the driver gene mutations in differentiated thyroid cancer contribute to the activation of the MAPK and PI3K/Akt pathways downstream of tyrosine kinase receptors such as RET and NTRKs. RLTs activate the MAPK and PI3K pathways, and CRAF induces negative feedback to inhibit MAPK. As a result, the anti-apoptotic effect via the PI3K pathway is predominant in RLTs. BLTs activate the MAPK pathway, which is highly proliferative. In well-differentiated thyroid tumors such as thyroid follicular nodular disease (or functional nodule), activation of the TSH receptor-mediated cAMP pathway is predominant; this promotes hormonal functions, such as iodine metabolism and the expression of hormone-related genes.

**Figure 2 biomedicines-12-00712-f002:**
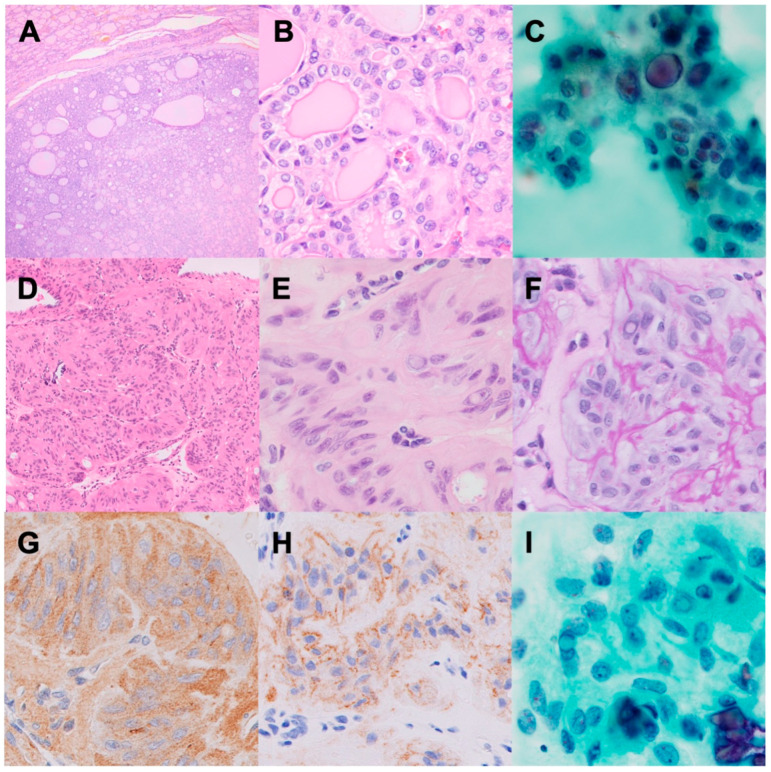
Low-risk neoplasms of the thyroid. (**A**–**C**): Non-invasive follicular thyroid neoplasm with papillary-like nuclear features (NIFTP). HE staining images of NIFTP at a low (**A**) and high magnification (**B**). Papanicolaou staining image of an NIFTP fine-needle aspiration specimen (**C**). (**D**–**I**): Hyalinizing trabecular tumor (HTT). HE staining images of HTT at a low (**A**) and high magnification (**B**). PAS staining image of HTT (**F**). Immunohistochemical staining with the MIB1 antibody revealed cytoplasmic (**G**) and cell membranous (**H**) immunoreactivity. Papanicolaou staining image of an HTT fine-needle aspiration specimen (**I**).

**Figure 3 biomedicines-12-00712-f003:**
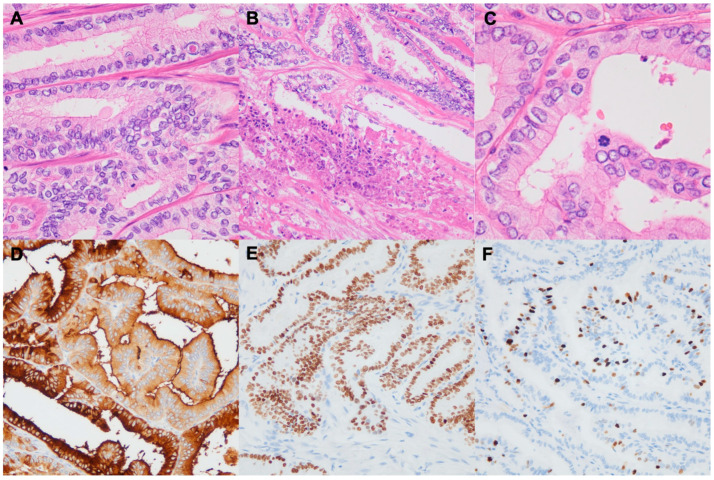
Differentiated high-grade thyroid carcinoma (DHGTC). HE staining images of a DHGTC case (**A**–**C**). This case is a tall-cell papillary thyroid carcinoma (PTC) (**A**) that meets the criteria of DHGTC. It shows tumor necrosis (**B**) and mitosis (**C**). Immunohistochemistry of Thyroglobulin (**D**) and TTF1 (**E**). The Ki-67 (MIB1) labeling index is about 15% (**F**).

**Figure 4 biomedicines-12-00712-f004:**
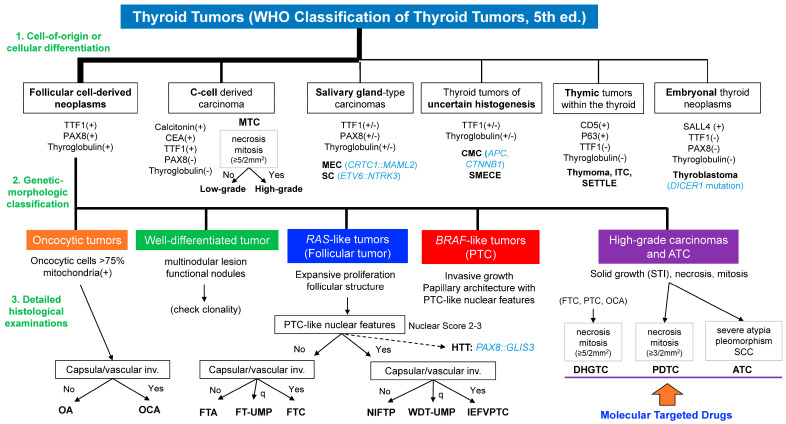
Algorithm for diagnosis of thyroid tumors (WHO 5th edition). The WHO 5th simplified the diagnosis of thyroid cancer by dividing it into three steps: (i) a consideration of the origin or cellular differentiation of the tumor cells, (ii) an assessment of the gene mutations, and (iii) a detailed examination of various histomorphological features, including capsular/vascular invasion, mitosis, and tumor necrosis. The cell of origin or cellular differentiation can be examined using immunohistological markers such as TTF1, PAX8, and calcitonin. The most frequent follicular cell-derived neoplasms are classified into five categories according to genetic mutations, which are closely related to tumor morphology. Oncocytic tumors and *RAS*-like tumors (RLTs) are commonly encapsulated and are further classified morphologically by the presence or absence of capsular/vascular invasion. Tumors with “Questionable (q)” invasion are diagnosed as FT-UMP or WDT-UMP. In RLTs, papillary-like nuclear features are also important. *BRAF* p.V600E-like tumors (BLTs) commonly show infiltrative growth with apparent PTC-like nuclear features and papillary structures. Tumors with predominantly solid growth are likely classified as high-grade carcinomas or anaplastic thyroid carcinomas (ATCs). Mitotic counts and tumor necrosis are critical in the diagnosis of high-grade carcinomas.

**Table 1 biomedicines-12-00712-t001:** Summary of the WHO classification, 5th edition (beta, 2022).

WHO 4th Edition	WHO 5th Edition
Follicular thyroid adenoma (FTA) Hyalinizing trabecular tumour (HTT) Other encapsulated follicular-patterned thyroid tumours Follicular tumour of uncertain malignant potential (FT-UMP), Well-differentiated tumour of uncertain malignant potential (WDT-UMP), Non-invasive follicular thyroid neoplasm with papillary-like nuclear features (NIFTP) Papillary thyroid carcinoma (PTC) Follicular thyroid carcinoma (FTC), NOS Hürthle cell tumours Hürthle cell adenoma, Hürthle cell carcinoma Poorly differentiated thyroid carcinoma (PDTC) Anaplastic thyroid carcinoma (ATC) Squamous cell carcinoma (SCC) Medullary thyroid carcinoma (MTC) Mixed medullary and follicular thyroid carcinoma Mucoepidermoid carcinoma (MEC) Sclerosing mucoepidermoid carcinoma with eosinophilia (SMECE) Mucinous carcinoma Ectopic thymoma Spindle epithelial tumor with thymus-like elements (SETTLE) Intrathyroidal thymic carcinoma (ITC)	1. Developmental abnormalities 2. Follicular cell-derived neoplasms 2.1 Benign tumours: Thyroid follicular nodular disease (TFND), FTA, Oncocytic adenoma (OA) 2.2 Low-risk neoplasm: NIFTP, Thyroid tumor UMP (TT-UMP), HTT 2.3 Malignant neoplasms: FTC, Invasive encapsulated follicular variant of papillary thyroid carcinoma (IEFVPTC), PTC, Oncocytic carcinoma (OCA), Differentiated high-grade thyroid carcinoma (DHGTC)/PDTC, ATC 3. Thyroid C-cell derived carcinoma MTC: low-grade, high-grade 4. Mixed medullary and follicular cell-derived carcinomas 5. Salivary gland-type carcinomas of the thyroid MEC, Secretory carcinoma 6. Thyroid tumours of uncertain histogenesis SMECE, Cribriform morular carcinoma 7. Thymic tumours within the thyroid Thymoma, SETTLE, ITC 8. Embryonal thyroid neoplasms Thyroblastoma

**Table 2 biomedicines-12-00712-t002:** Papillary thyroid carcinoma (PTC) subtypes (WHO 5th edition, 2022).

PTC Subtypes	Propotion Required for Diagnosis	Major Genetic Abnormalities	Prognosis
Classic PTC	-	*BRAF* p.V600E, *CCDC6::RET, NCOA4::RET*	-
Infiltrative follicular PTC	≥90%	*BRAF* p.V600E, *NRAS, RET* fusions, *NTRK* and *ALK* fusions	favorable
Tall cell PTC	≥30%	*BRAF* p.V600E, miR-21, *TERT* promoter, *TP53*	high risk
Columnar cell PTC	Not available (NA)	*BRAF* p.V600E, *BRAF* fusions, *CDKN2A* loss, gain of ch.1	high risk
Hobnail PTC	≥30%	*BRAF* p.V600E, *TERT* promoter, *TP53, PIK3CA*	high risk
Solid/trabecular PTC	>50	*CCDC6::RET, NCOA4::RET, ETV6::NTRK3, BRAF* p.V600E (rare)	Slightly higher risk
Diffuse sclerosing PTC	100%	*NCOA4::RET, BRAF* p.V600E(20%), *ALK* fusions (10%)	Slightly higher risk
PTC with fibromatosis/fasciitis like/desmoid-type stroma	-	*CTNNB1* mutations, *BRAF* p.V600E (frequent)	unknown
Oncocytic PTC	NA (>75%?)	*BRAF* p.V600E, *GRIM-19, RET* fusions	Equivalent to classic
Warthin-like PTC	NA	BRAF p.V600E	unknown

**Table 3 biomedicines-12-00712-t003:** Diagnostic criteria for high-grade follicular cell-derived non-anaplastic thyroid carcinoma and high-grade medullary carcinoma.

	Poorly Differentiated Carcinoma (PDTC)	Differentiated High-grade Thyroid Carcinoma (DHGTC)	Medullary Carcinoma (MTC), High-Grade
Cell-of-origin	Follicular cell	Follicular cell	C cell
Growth pattern	Solid/trabecular/insular	Papillary, follicular, (solid)	Any
Nuclear findings	No PTC-like nuclear atypia	Any	Any
Other conditions	(At least one of the following features is satisfied)		
	1. Mitosis (≥3/2 mm^2^) 2. Tumor necrosis 3. Convoluted nuclei	1. Mitosis (≥5/2 mm^2^) 2. Tumor necrosis	1. Mitosis (≥5/2 mm^2^) 2. MIB1 labeling idex ≥5% 3. Tumor necrosis
Anaplastic features	None	None	None

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
