# Peer review of "Molecular Pathology of Thyroid Tumors: Essential Points to Comprehend Regarding the Latest WHO Classification"

_biomedicines, 2024, doi:10.3390/biomedicines12040712_

Round 1

Reviewer 1 Report

Comments and Suggestions for Authors

The paper is very timely and focuses on the hot-topics of the recent WHO Classification of Thyroid Neoplasms.

Things to manage:

-          Revision of the framework for benign lesions: oncocytic adenoma with its genetic background is missing (it should be discussed at this point and not afterwards).

-          Low-risk tumors: Please be more careful in picturing genetic background of NIFTP, FT-UMP, WDT-UMP, which belong to the RLTs.

-          Subtypes of PTC: Please state the difference between infiltrative and invasive.

-          Diagnostic procedure according to WHO 5th: please expand in order to provide a comprehensive view of what described in Figure 4.

-          Figure 4: well report the high grade tumors by providing a chart differentiating PDTC-DHGTC-ATC.

-          Please provide a separate paragraph where the description of the main mutations and the histology counterparts are reported. Please cite: Benign thyroid nodules with RAS mutation grow faster. Clin Endocrinol (Oxf). 2015 Aug 11. doi: 10.1111/cen.12875. [Epub ahead of print] PubMed PMID: 26260959; BRAF mutation in cytology samples as a diagnostic tool for papillary thyroid carcinoma. Expert Opinion on Medical Diagnostics Jul 2011, Vol. 5(4): 277-290.

-          Please better address the role of the concomitant mutations, especially referring to the TERT/BRAF co-occurrence (please cite  Application of molecular biology of differentiated thyroid cancer for clinical prognostication. Endocr Relat Cancer. 2016 Aug 30. pii: ERC-16-0372. [Epub ahead of print] PubMed PMID: 27578827.).

Comments on the Quality of English Language

.

Author Response

I deeply appreciate the useful comments from reviewer 1. Based on these comments, I revised the manuscript as follows.

Things to manage:

-          Revision of the framework for benign lesions: oncocytic adenoma with its genetic background is missing (it should be discussed at this point and not afterwards).

I agree with the suggestion.  Oncocytic adenoma with its genetic backgrounds are added to the framework for benign lesions.

-          Low-risk tumors: Please be more careful in picturing genetic background of NIFTP, FT-UMP, WDT-UMP, which belong to the RLTs.

I agree with the comments.  I removed TT-UMP from the Figure 1.  I also added following descriptions of the genetic backgrounds of NIFTP, FT-UMP, and WDT-UMP in the “Low-risk tumors” section.

“Driver gene mutations in TT-UMP are diverse. While RAS-like mutations are the most common type, mutations in EIF1AX and TSHR, as well as PAX8::PPARG rearrangements, can also be detected.”

-          Subtypes of PTC: Please state the difference between infiltrative and invasive.

I appreciate and agree with the comments. “Infiltrative” and “invasive” are the same. Unfortunately, the WHO 5th does not standardize these terms. I used the term “invasive” for description except for the histological name defined in the WHO 5th (line 124).

-          Diagnostic procedure according to WHO 5th: please expand in order to provide a comprehensive view of what described in Figure 4.

I agree with the comments. I added a detailed description of the diagnostic procedure section in the revised manuscript (p.9, lines 315-p.11, line 411).

-          Figure 4: well report the high grade tumors by providing a chart differentiating PDTC-DHGTC-ATC.

I revised the Figure 4.

-          Please provide a separate paragraph where the description of the main mutations and the histology counterparts are reported. Please cite: Benign thyroid nodules with RAS mutation grow faster. Clin Endocrinol (Oxf). 2015 Aug 11. doi: 10.1111/cen.12875. [Epub ahead of print] PubMed PMID: 26260959; BRAF mutation in cytology samples as a diagnostic tool for papillary thyroid carcinoma. Expert Opinion on Medical Diagnostics Jul 2011, Vol. 5(4): 277-290.

Thank you very much for suggesting the references. I cited them and revised section 2.1. I added the following description of the main mutations for each histological counterpart (lines 76-85).

BRAF p.V600E mutation, along with RET and NTRK rearrangements, have been identified as primary genetic aberrations in PTC [2-4].  Notably, BRAF p.V600E exhibits a high specificity for PTC.  Their detection, even in minute quantities within fine needle aspiration cytology specimens, plays a pivotal role in diagnosing PTC [6].  Additionally, RAS point mutations have been documented in follicular PTC and FTC.  It's worth noting that RAS mutations have also been observed in FTA and benign thyroid nodules.  Furthermore, nodules harboring RAS mutations have been associated with accelerated growth rates [7].  Unlike the well-known BRAF p.V600E mutation, other alterations in the BRAF gene, such as p.K601 mutations and gene rearrangements, have been detected in FTC as well as benign follicular tumors [2-4].”

-          Please better address the role of the concomitant mutations, especially referring to the TERT/BRAF co-occurrence (please cite  Application of molecular biology of differentiated thyroid cancer for clinical prognostication. Endocr Relat Cancer. 2016 Aug 30. pii: ERC-16-0372. [Epub ahead of print] PubMed PMID: 27578827.).

I added the following description regarding PTC's “Genetic Duet” by citing the references (lines 252-257).

“Several studies suggested the prognostic importance of co-existence of BRAF p.V600E and TERT promoter mutations in differentiated thyroid cancer [20,21].  This combination of BRAF p.V600E and TERT promoter mutations was called a “genetic duet”.  Recent evidence further recognized the importance of regulatory single nucleotide polymorphism (rSNP), rs2853669, in the TERT promoter as an additional risk predictor for PTC [22,23].”

Reviewer 2 Report

Comments and Suggestions for Authors

Comments 

1. Author may give recent epidemiology of Thyroid Ca.

2. In table 1, author should compare the WHO 4thand 5th revisions and may combine with table-2. 

3. section 2.7. other tumor- means ?. Author may give an appropriate heading to this section. 

4. English language and grammar need to be correct.

Comments on the Quality of English Language

Minor editing of English language required

Author Response

I deeply appreciate useful comments from reviewer 2.  Based on the comments, I revised the manuscript as follows.

Comments 

  1. Author may give recent epidemiology of Thyroid Ca.

I added the following description of the recent epidemiology of thyroid cancer in the introduction.

“Thyroid cancer diagnoses, particularly in women, are on the rise.  Improved detection methods likely contribute to this trend, although environmental factors are also under investigation.  Despite the increase in diagnoses, mortality rates remain low.”

  1. In table 1, author should compare the WHO 4thand 5th revisions and may combine with table-2. 

According to the comment, I revised Table 1, which now contains the information of WHO 4th.  I did not combine Tables 1 and 2 because they looked much more complicated.

  1. section 2.7. other tumor- means ?. Author may give an appropriate heading to this section. 

Thank you for the comments. I renamed it as “Miscellaneous thyroid tumors” in the revised version.

  1. English language and grammar need to be correct.

I apologize for the inconvenience in reviewing the manuscript. The native speaker checked the manuscript and revised it.

Round 2

Reviewer 1 Report

Comments and Suggestions for Authors

The paper has been improved as requested.